# Storm surge and ponding explain mangrove dieback in southwest Florida following Hurricane Irma

David Lagomasino [1✉], Temilola Fatoyinbo [2], Edward Castañeda-Moya [3], Bruce D. Cook[2], Paul M. Montesano[2,4], Christopher S. R. Neigh [2], Lawrence A. Corp[2,4], Lesley E. Ott[2], Selena Chavez [5] & Douglas C. Morton[2]

Mangroves buffer inland ecosystems from hurricane winds and storm surge. However, their ability to withstand harsh cyclone conditions depends on plant resilience traits and geomorphology. Using airborne lidar and satellite imagery collected before and after Hurricane Irma, we estimated that 62% of mangroves in southwest Florida suffered canopy damage, with largest impacts in tall forests (>10 m). Mangroves on well-drained sites (83%) resprouted new leaves within one year after the storm. By contrast, in poorly-drained inland sites, we detected one of the largest mangrove diebacks on record (10,760 ha), triggered by Irma. We found evidence that the combination of low elevation (median = 9.4 cm asl), storm surge water levels (>1.4 m above the ground surface), and hydrologic isolation drove coastal forest vulnerability and were independent of tree height or wind exposure. Our results indicated that storm surge and ponding caused dieback, not wind. Tidal restoration and hydrologic management in these vulnerable, low-lying coastal areas can reduce mangrove mortality and improve resilience to future cyclones.

[1] Department of Coastal Studies, East Carolina University, Wanchese, NC, USA. [2] Biospheric Sciences Laboratory, NASA Goddard Space Flight Center, Greenbelt, MD, USA. [3] Institute of Environment, Florida International University, Miami, FL, USA. [4] Science Systems and Applications, Inc., Lanham, MD, USA. [5] Department of Earth and Environment, Florida International University, Miami, FL, USA. ✉email: lagomasinod19@ecu.edu

Worldwide, mangrove forests safeguard inland areas from coastal storms[1,2]. In the US alone, mangroves prevent $11.3 billion in property damage and 14,200 km² of flooding each year, with the greatest flood protection benefits during cyclones[2]. Coastlines with extensive mangrove forests also buffer coastal economies by reducing the period of economic inactivity following a hurricane by up to four months when compared to areas with minimal mangrove cover[1,3]. Projected changes in the frequency and intensity of tropical cyclones[4,5] may create a positive feedback of forest loss whereby more frequent mangrove damages from tropical cyclones compromise the buffering capacity of mangroves in future storms[6].

Damage to mangrove forests from tropical cyclones varies from temporary defoliation to widespread tree mortality[6,7]. Subsequent recovery from storm damage also varies as a function of storm strength and edaphic conditions. In some cases, storms deposit phosphorus-rich sediments that stimulate mangrove growth[8]. However, in other areas, initial damages are compounded by delayed mortality that can limit mangrove recovery for months or years following a storm[9,10]. Identifying the locations and the mechanisms that trigger widespread collapse of mangrove forests is therefore critical for understanding the vulnerability of coastal ecosystems and developing plans to mitigate dieback events from future storms[11].

South Florida is home to the largest tract of continuous mangrove forests in the United States, of which 75% (144,447 ha) of the country's mangroves occur within Everglades National Park alone[12]. Human development has hemmed in much of the remaining mangroves, limiting landward migration, and altered coastal hydrology, increasing vulnerability to sea level rise, salt water intrusion, and ponding[13,14]. These chronic stressors are compounded by strong and sustained winds, storm surge, and prolonged flooding during hurricane events, pushing mangroves to the brink of collapse[7,10]. Spatial variability in the risk of mangrove dieback depends on how specific characteristics of each hurricane[15] interact with forest structure, species composition, geomorphology, and elevation[8,10], and prior hydrologic connectivity[16].

In September 2017, Hurricane Irma made landfall in south Florida with winds in excess of 52 mps (116 mph) and storm surge as high as 3 m (Fig. 1). Mangroves along the southwest coast experienced the full strength of the storm. Powerful winds stripped leaves and branches from mangroves and snapped and uprooted trees. Storm surge reshaped coastal topography through sedimentation, erosion, and inundation of low-lying areas. At the local-scale, structural damages to mangroves and the reorganization of coastal geomorphology have been shown to threaten the long-term stability of the ecosystem by altering drainage patterns and disrupting forest succession[17,18]. Previous studies have used satellite imagery across small geographic extents to track the damage and recovery of mangroves from hurricanes[15,19], but few studies have assessed the three-dimensional (3D) changes in mangrove structure[20] that are needed to understand the diversity of hurricane impacts on mangrove forests and the limits to mangrove resilience at a large scale.

Here, we combine airborne and satellite remote sensing data to estimate mangrove damage and recovery in the year following Hurricane Irma (Fig. 1). We analyze airborne lidar data collected before (April 2017) and after (December 2017) the storm with NASA Goddard's Lidar, Hyperspectral, and Thermal (G-LiHT) airborne imager[21] to estimate the 3D changes in vegetation structure at 1-m spatial resolution across 130,000 ha of coastal wetlands in south Florida (Fig. 1, Supplementary Fig. 1). We combine the G-LiHT data with high-resolution satellite stereo imagery and Landsat time series information to track the recovery of coastal ecosystems across gradients of exposure to maximum hurricane winds, storm surge, community composition, and ground elevation (Supplementary Figs. 1–8) (Materials and methods are available as Supplementary materials). By intersecting the measured structural damage and recovery trajectories with species composition maps and topographic elevation models, we quantify how hurricane winds damage mangrove forests, reducing canopy height and fractional vegetation cover. However, storm surge, elevation, and landscape position determine the trajectory of forest recovery following initial damages, and these factors are the main drivers of long-term dieback. Together, these data capture the spatial and temporal patterns of mangrove damage and recovery following a major hurricane and underpin recommendations to monitor and address coastal vulnerability in hurricane-prone regions.

## Results

Hurricane Irma triggered one of the largest recorded mangrove dieback events in the region. In the first 15 months after Irma, 10,760 ha of mangroves showed evidence of complete dieback, with little to no greening (Fig. 2, Supplementary Fig. 3). These low resilience areas were marked by a 0.2 drop in NDVI and recovery times that exceeded 15 years (Materials and methods are available as Supplementary materials). The areas of dieback primarily occurred in large concentrated patches at low elevations (Fig. 2a). Indeed, the majority of the dieback, or low resilience areas, occurred where ground elevation was under 10 cm asl (Fig. 2b, Supplementary Fig. 4, Supplementary Table 1). The largest mangrove die-off region occurred along the southern tip of Florida in a location that is separated from the ocean by the Buttonwood Ridge, a natural coastal barrier/depositional berm (~0.5 m asl in height) that limits tidal exchange, impounds water, and isolates inland mangroves from the direct hydrological exchange with Florida Bay[22] (Fig. 2). Similar artificial barriers (e.g., roads and levees) are also present in the region that limit inflow of freshwater from upstream sources. In addition, our analysis found other large patches of mangrove die off around Gopher Key and Ten Thousand Islands, areas that are also semi-enclosed by coastal berms.

The maximum recorded storm surge level of ~3 m impacted the southwestern coast but was less than 2 m on the southern coast (Fig. 1). However, over 90% of the dieback occurred in areas where storm surge exceeded 1.4 m above the ground surface (Fig. 2c, Supplementary Fig. 4). Dieback was also disproportionately concentrated in forest stands dominated by *Avicennia germinans* with an estimated impact area of 73% (7,750 ha) of all dieback areas (Fig. 2d). Overall, Irma had a strong selective pressure on the distribution of forests dominated by *A. germinans* whereby nearly 40% of these mangrove forests died compared to the less than 6% of the other forest communities.

Where mortality and recovery were closely related to soil elevation and storm surge, structural damages to mangrove forests caused by Hurricane Irma, as measured by the change in satellite and lidar canopy height models, varied based on wind exposure and pre-storm canopy height (Fig. 1, Supplementary Fig. 5-6). Strong and sustained winds reduced average canopy height by 1.16 m (S.D. ± 1.36) (Fig. 3, Supplementary Table 2). In forests with mean heights over 10 m, canopy height decreased by an average of 2.05–2.97 m, regardless of hurricane wind speed, whereas shorter trees suffered smaller losses (<1.2 m). These reductions in canopy height reflect the loss of canopy branches and snapped or uprooted whole trees (Fig. 1). Combining coincident high-resolution imagery from airborne and satellite platforms with observed canopy height losses (Materials and methods are available as Supplementary materials), we estimated a 15.3% (±10.6%) reduction in mangrove canopy volume, a measure

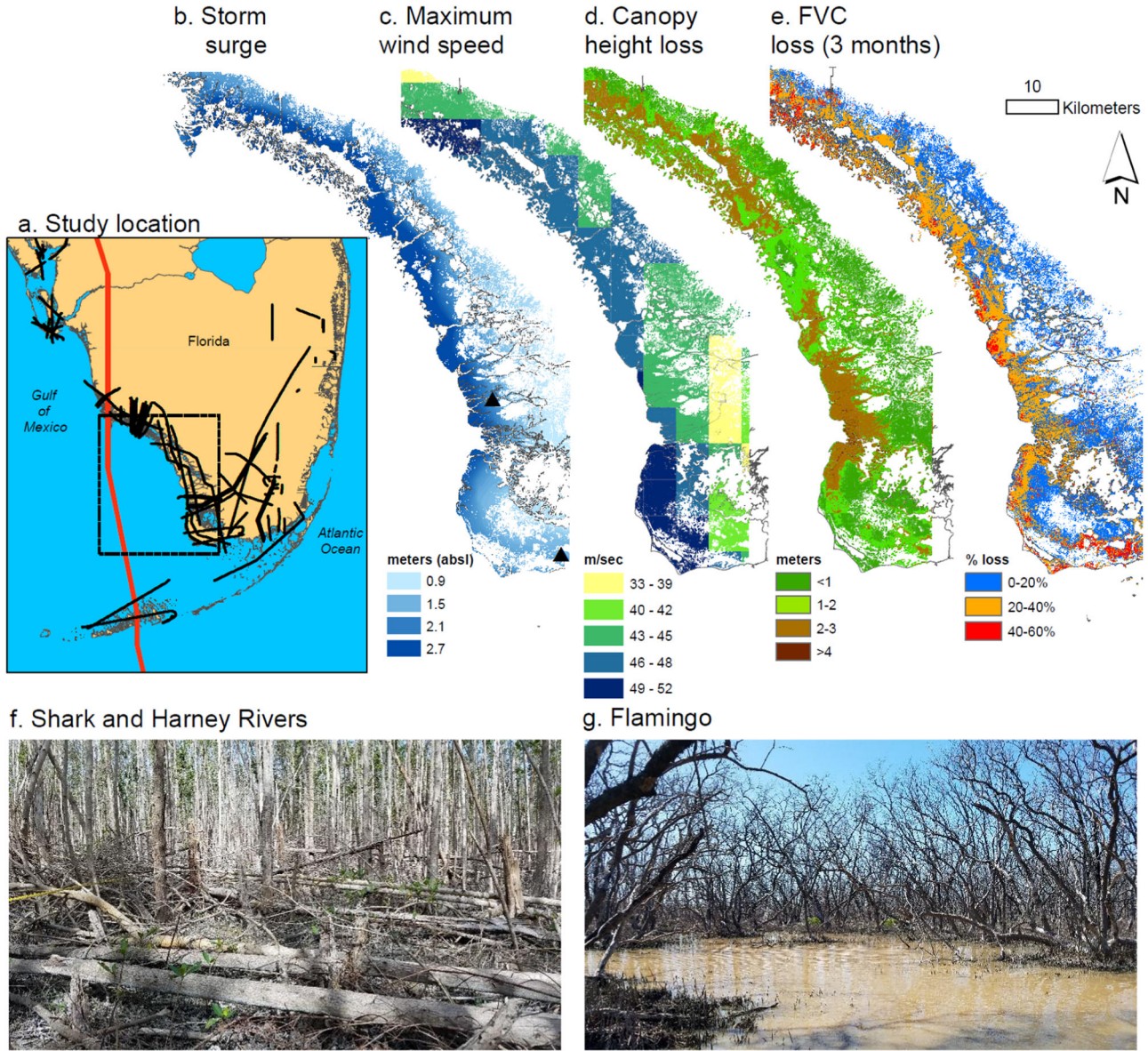

**Fig. 1 Tall mangroves in southwest Florida experienced the greatest canopy height losses from strong winds but storm surge and hydrologic barriers led to mangrove dieback in poorly-drained semi-enclosed inland areas for mangroves in all forest height classes. a** Track of Hurricane Irma is highlighted in red. NASA G-LiHT lidar coverage is outlined in black. **b** Modeled storm surge from the Coastal Emergency Risk Assessment. **c** Maximum hourly wind speeds during Hurricane Irma from the Goddard Earth Observing System, Version 5 model. **d** Estimated canopy height losses using airborne lidar and high-resolution satellite stereo imagery. **e** Loss of fractional vegetation cover (FVC) after Hurricane Irma using pre- and post-hurricane Landsat imagery. **f** Photo collected January 2018 in lower Harney River Estuary. **g** Photo collected December 2018 in Flamingo. Photo locations are shown in (**b**) as triangles.

strongly correlated with aboveground biomass[23]. The largest losses of canopy height and volume were concentrated in the major estuaries (e.g., Shark and Harney Rivers) in the southwestern Everglades, which were also areas with the tallest forest canopies in the region before Irma and regular freshwater input and tidal flushing (Fig. 1d and f). On average, forested areas with high resilience had shorter canopies (<6.2 m) prior to the hurricane and also experienced smaller canopy height losses than regions with slower recovery (Fig. 3, Supplementary Fig. 4).

Hurricane Irma also led to widespread mangrove canopy defoliation, measured as a loss of fractional vegetation cover (FVC) directly after the storm. Irma reduced the extent of closed-canopy mangrove forests by 86%. Importantly, areas of greatest FVC loss were not co-located with the largest reductions in canopy height, except where Irma made landfall (Fig. 1,

Supplementary Fig. 7). Losses in canopy cover were widespread across the storm-affected area, whereas canopy height losses were primarily confined to areas with the tallest trees. Areas with >40% decline in FVC (6180 ha) were more likely to experience mangrove dieback (61.5%), in contrast to the 38.5% of mangroves with severe canopy cover losses that were classified as intermediate and high resilience areas (Supplementary Fig. 8).

## Discussion

The spatial and temporal patterns of mangrove forest damage and recovery in Florida following Hurricane Irma highlight how storm surge, position in the tidal frame, and drainage drove mangrove dieback (Fig. 4). Delayed or failed mangrove recovery was primarily confined to low elevation, endorheic basins, and

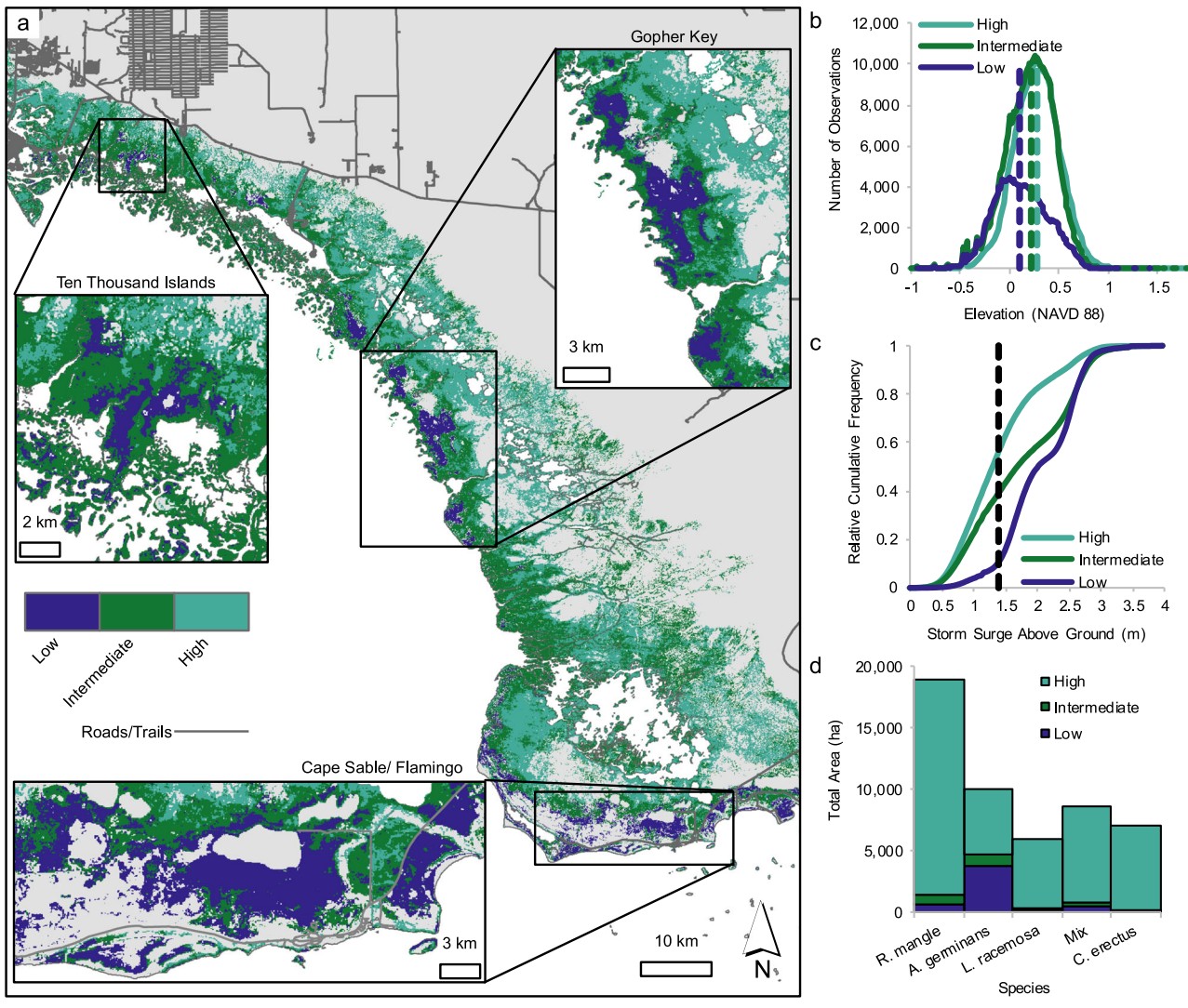

**Fig. 2 Mangrove forest dieback was concentrated in low-lying areas that are typically dominated by *A. germinans*, the most salt-tolerant species in the neotropics.** Hotspots of dieback are highlighted for Ten Thousand Islands, Gopher Key, and Cape Sable/ Flamingo. **a** Distribution of resilience class across southwest Florida. **b** Distribution of ground elevation for each resilience type. Dashed lines indicate the median elevation values per class. **c** Cumulative frequency of storm surge above ground by resilience class. **d** Area of resilience class by dominant mangrove species. Results of the Kolmogorov–Smirnov goodness-of-fit can be found in Supplementary Fig. 4 and Supplementary Table 1.

| | Maximum Wind Speed (m sec⁻¹) | | | | | | |
|---|---|---|---|---|---|---|---|
| Canopy Height (m) | 25-30 | 30-35 | 35-40 | 40-45 | 45-50 | >50 | Average |
| >20 m | -1.36 | -1.6 | -2.83 | -2.91 | -3.3 | -3.96 | -2.97[a] |
| 15-20 m | -3.07 | -2.1 | -2.18 | -2.41 | -2.57 | -2.72 | -2.49[b] |
| 10-15 m | -2.37 | -1.98 | -1.72 | -2.3 | -2.01 | -1.1 | -2.05[c] |
| 5-10 m | -1.73 | -1.65 | -0.95 | -1.48 | -1.1 | -0.73 | -1.18[d] |
| 0-5 m | -0.75 | -0.5 | -0.28 | -0.42 | -0.47 | -0.54 | -0.41[e] |
| Average | -1.55 | -1.41 | -0.58 | -1.41 | -1.25 | -0.63 | -1.16 |

Estimated Mangrove Area Covered

- No Data - Modeled Values
- <60 ha
- 60-800 ha
- 801-1500 ha
- 1501-2500 ha
- >2501 ha

**Fig. 3 Changes in average canopy height, in meters, from Hurricane Irma were greatest in taller mangrove forests.** Wind speed classes reflect the maximum one-hour wind speed exposure from Hurricane Irma (Materials and methods are available as Supplementary materials). Colors indicate the mangrove area covered by each height and wind class. Lettered superscripts denote classes that are statistically significant ($p < 0.00001$) using a one-way ANOVA paired with a post-hoc Tukey test. Additional information on the standard errors and area estimates can be found in Supplementary Table 2.

interior portions of larger mangrove forest patches. Impounded floodwaters cut off from the tidal prism can lead to multiple biochemical stressors, particularly increases in porewater sulfides, a phytotoxin in wetlands that accumulates during permanent flooding conditions[24]. Artificial barriers such as roads and levees can obstruct the flow of water and exacerbate ponding and flooding conditions, leading to mangrove die-off. Similarly, natural shoreline embankments formed by sediment deposited during minor storms can also restrict drainage, increasing the susceptibility to impoundment from hurricane storm surge, as

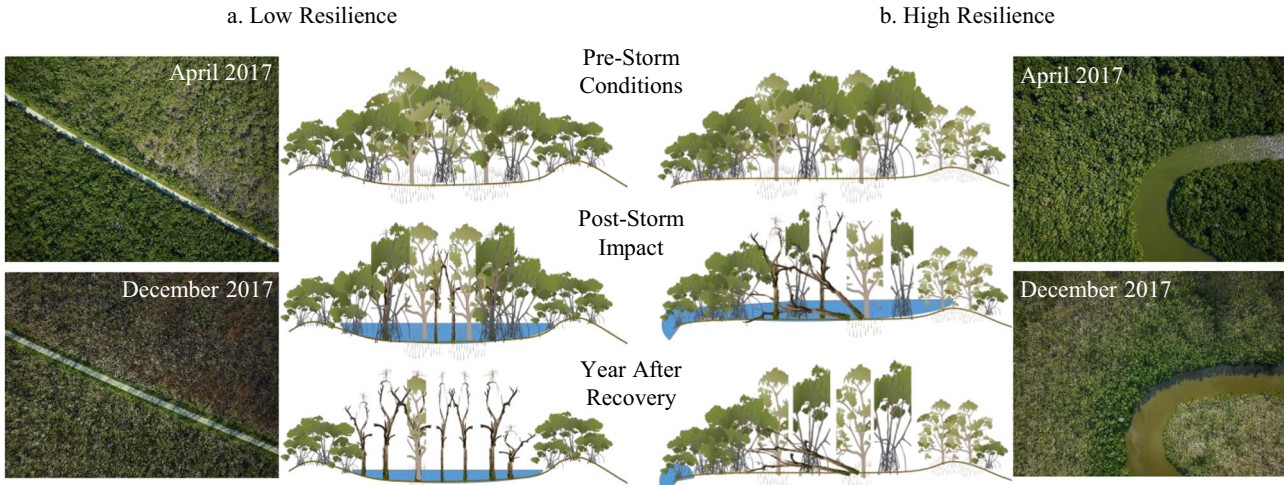

**Fig. 4 Forest canopy structure, topographic position, and drainage contribute to differences in mangrove vulnerability and resilience to hurricane damages. a** Areas that flood from salt water storm surge but do not drain have low resilience, with greatest risk for dieback from hypersalinization and pore water sulfide build up. **b** Mangroves in well-drained (i.e., tidally dominated) sites have high resilience to storm damages and can flush new leaves or resprout quickly after canopy damage from hurricane winds. Aerial photos from NASA's G-LiHT Airborne Imager show examples of mangroves with low and high-resilience in southwest Florida before and after Hurricane Irma.

seen near Gopher Key and Cape Sable (Figs. 2c, 4a). Notably, these two regions accounted for nearly 75% of the total dieback, despite not being in the direct path of the storm. Hydrologic isolation in these areas may have stressed or weakened mangroves prior to Irma, but time series of satellite data in this study confirmed synchronized and widespread mortality following hurricane wind and flood damage (Supplementary Fig. 3).

Pre-storm forest structural attributes (e.g., canopy height, fractional vegetation cover), species composition, and storm characteristics further modified the patterns of initial mangrove forest damage and the timescales of recovery[9,17]. Taller (>10 m) trees suffered greater damages from exposure to hurricane winds. The height dependence for mangrove canopy damage in this study is consistent with similar relationships reported in other forests across the Caribbean region[25]. Yet, branch loss and stem damage to tall mangroves did not trigger dieback in these forest areas. At least two factors contributed to differences in the spatial patterns of damage from wind and storm surge. First, taller forests were most common in the Shark and Harney Rivers, areas with tidal connectivity and drainage that limited ponding and the risk of hypersalinization. Second, these mangrove areas with tidal connectivity have high soil nutrient fertility as a result of phosphorus-rich mineral sediments deposited by hurricanes that facilitate canopy refoliation and seedlings growth[8], two mechanisms that contributed to rapid recovery of FVC in these areas despite large initial canopy height losses. Plant traits also influence how canopy or stem damage impacts forest recovery, including the potential for a shift in species composition following hurricane damages. For example, *Rhizophora mangle* does not resprout after stem damage, leading to higher rates of mortality in this species from direct hurricane wind impacts than *A. germinans* or *Laguncularia racemosa*[17]. Whether differential sensitivity to hurricane winds results in a long-term shift in mangrove forests merits further study.

We found greater vulnerability of monospecific and co-dominated stands of *A. germinans* to dieback following Hurricane Irma than other species communities. However, this species-specific vulnerability may reflect the competitive exclusion of other species in low-lying or hydrologically isolated areas prior to the storm, as these basin mangrove ecotypes are generally dominated or co-dominated by *A. germinans*[26], the most salt-

tolerant species in the neotropics. Prolonged exposure to salt water following Irma triggered widespread mortality, overwhelming even the ability of *A. germinans* to withstand longer periods of inundation than *R. mangle* or *L. racemosa*. It is also possible that stressful conditions in basin mangrove areas predisposed these stands to dieback. Excessive ponding reduces primary productivity[27] and suppresses nutrient uptake, and prolonged exposure to inundation can create toxic conditions for mangroves and accelerate the degradation of fine root material and soil substrate, ultimately leading to peat collapse[28].

The south-to-north track of Hurricane Irma resulted in greater mangrove dieback than other recent hurricanes to strike southwest Florida, including Category 4 or Category 5 Hurricanes Charley (2005) and Andrew (1992), respectively[17,29]. The last two major storms with a similar trajectory over the Florida Keys and Florida Bay, Hurricane Donna (1960, Category 4) and the Labor Day Hurricane (1935, Category 5)[30], also triggered widespread damage and the eventual collapse of coastal forests[18]. Unlike Irma, little is known about the total extent of dieback from Donna or the Labor Day Hurricane. Yet, this storm trajectory exposes some of the most vulnerable, low-lying coastal endorheic landscapes to the strongest hurricane winds and wind-driven storm surge[31]. High storm surge from Hurricane Irma overtopped the Buttonwood Ridge, a relatively high natural coastal barrier[32], and became trapped (Fig. 4). Dieback from Irma occurred in areas with even lower storm surge (1.4 m above ground), highlighting the importance of both hurricane strength and the path of the storm directing wind damage and storm surge on mangrove forests of southwest Florida.

In this study, we identified patterns of mangrove forest dieback associated with a combination of environmental attributes associated with vulnerability such as topographic position, forest structure, and species composition, and storm characteristics. Increasing vulnerability from sea level rise, hydrologic isolation, or drought lowers the threshold for mangrove dieback events following extreme events, such as hurricanes[33]. In some cases, these individual agents could even push ecosystems to the point of collapse (or state change) even in the absence of an extreme event, as in the case of salt water intrusion[34], which can dramatically shift communities from wetland to aquatic systems with associated changes in biogeochemical cycling[35]. Tropical cyclones

are one of the major causes of reported global mangrove mortality[36] and increasing storm frequency or intensity, even in the absence of increased vulnerability, may also accelerate forest dieback and associated shifts in species composition and coastal protection[37].

Extreme weather events account for 11% of global mangrove forest loss since the start of the 21st century[38], highlighting how the combined impacts of tropical cyclones, heat waves, and droughts drastically alter coastal ecosystem structure and function[25,39]. Sea level rise and the projected increase in the frequency and severity of hurricanes with climate change[40] provide strong motivation to augment traditional hurricane rating systems with specific metrics that account for storm surge and coastal geomorphology. In addition, integrated coastal monitoring networks can enhance the efficacy of surge warnings and mitigate future mangrove dieback events to protect both human and natural systems. For example, adding field measurements in vulnerable, low-lying areas to ongoing long-term monitoring networks in more pristine sites can help identify key physical and biological processes[41]. Regular satellite monitoring and frequent coastal lidar surveys can locate endorheic basins and indicators of ecosystem stress to help inform risk assessments and prioritize rehabilitation or removal of barriers needed to improve coastal resilience and trajectories of recovery post-disturbance. Remediation efforts to restore freshwater inputs and tidal regimes will help to reduce the stress of prolonged flooding in vulnerable basin mangrove forests. These monitoring and mitigation efforts to enhance coastal resilience to future storms will require cooperation across multiple water resource agencies to reduce vulnerability in coastal communities and minimize economic losses during extreme events[1–3].

## Methods

A combination of airborne and satellite remote sensing data were used to quantify changes in mangrove forest structure and function from Hurricane Irma (Supplementary Fig. 1). Findings based on multi-sensor airborne data were scaled to the entire study area using estimates of forest structure and vegetation phenology derived from satellite data.

**G-LiHT Airborne campaign**. During April 2017, NASA Goddard's Lidar, Hyperspectral, and Thermal (G-LiHT) airborne imager conducted an extensive airborne campaign in South Florida covering >130,000 ha. The same flight lines were resurveyed with G-LiHT eight months later, during November and December of 2017, to quantify structural changes in coastal forests of South Florida and Everglades National Park (ENP) following Hurricane Irma (Fig. 1). Lidar data was collected with two VQ-280i (Riegl USA) and synced during flight using RiAC-QUIRE version 2.3.7. The plane flew at a nominal height of 335 m above ground level at a pulse repetition frequency of 300 kHz to collect ~12 laser pulses per square meter. The analysis of pre- and post-hurricane conditions used 1-m resolution lidar data products (Supplementary Fig. 2) and 3-cm resolution stereo aerial and ground photos to estimate changes in vegetation structure, fractional cover, and terrain heights across the study domain. G-LiHT lidar canopy height models, digital terrain models, and estimates of fractional vegetation canopy cover (FVC) were produced using standard processing methodology[21]. All Level 1 through 3 lidar data products and fine-resolution imagery are openly shared through the G-LiHT webpage (https://gliht.gsfc.nasa.gov/).

**High resolution stereo maps of canopy height**. Stereo imagery from high-resolution commercial satellites can be used to estimate canopy and terrain surfaces[42,43]. Here, we derived digital surface models (DSMs) from DigitalGlobe's WorldView 2 Level 1B imagery. DigitalGlobe provides these data to U.S. Government agencies and non-profit organizations that support U.S. interests via the NextView license agreement[44]. The spatial resolution of these data depends on the degree of off-nadir pointing for each acquisition. In this study, image resolution ranged from 0.5 to 0.7 m. We selected along-track stereopairs within the study domain to identify stereo image strips (each ~17 km × 110 km) that were nominally cloud-free over the forested domain of interest for years 2012–2013, the most recent cloud-free stereo data available for the study region prior to Hurricane Irma. The DSMs were produced using the Ames Stereo Pipeline (ASP) v. 2.5.1 on the NASA Center for Climate Simulation's Advanced Data Analytics Platform at Goddard Space Flight Center (ADAPT, https://www.nccs.nasa.gov/services/adapt). The Worldview DSMs have been shown to accurately estimate mangrove canopy

height when compared to airborne lidar and radar interferometry[42,43]. The processing workflow was adapted from ref. [45], and was implemented semi-global matching algorithms with a $5 \times 5$ correlation kernel, and a $3 \times 3$ median-filter applied to the output point cloud prior to producing a 1 m DSM using a weighted average gridding rule[46]. The ASP processing yielded five DSMs at 1-m resolution that were used to capture pre-storm canopy surface elevations.

Each of the five Worldview DSMs were individually calibrated using overlapping pre-storm G-LiHT lidar data to estimate mangrove canopy heights across the study region (Supplementary Fig. 1). We sampled 1000 points within the mangrove forest cover (see mangrove classification, below) to develop a bias-correction equation between G-LiHT lidar-derived canopy heights and stereo DSM elevations (Supplementary Fig. 6). The bias-corrected canopy height models from high-resolution stereo imagery were mosaicked together to generate a 1-m resolution CHM for the entire study region (Supplementary Fig. 7). A pre-storm canopy volume was calculated by summing the 1 m × 1 m WorldView CHM for the entire region of interest. Similarly, a post-storm canopy volume was derived using the canopy damage model (see the section below), the relationship between the pre-storm CHM and the max wind speed. This analysis was conducted in ArcMap 10.7.1.

**Landsat mangrove forest classification**. Landsat 8 Operational Land Imager (OLI) imagery was used to map mangrove cover for the southern Florida study region. The imagery was preprocessed to surface reflectance[47] and clouds were masked following methods outlined in ref. [48]. The Surface Reflectance Tier 1 product in Google Earth Engine was used to create a cloud-free image mosaic for 2016 based on the median values of all cloud-free images for the year for all bands (Supplementary Fig. 1).

Training points were hand-selected using contemporary Google Earth imagery, field photos, and expert knowledge of the region. Twenty-four polygons covering a mangrove area of 1243 ha and 17 polygons covering a non-mangrove area of 2759 ha were identified for training regions. Within each of the two classes (i.e., mangrove and non-mangrove), 100,00 points were sampled and used for the training data in a Random Forest Classification implemented in Google Earth Engine[49]. The Random Forest model used 20 trees and a bag fraction of 0.5. The Landsat-based mangrove map was validated using the Region 3 species land cover map developed by the National Park Service for Everglades National Park[50]. The National Park species map was reclassified into mangrove and non-mangrove land cover, and 500 randomly generated points were sampled within each of the two land cover classes. The resulting error matrix indicated an overall accuracy of 90.6%.

**Post-storm canopy cover**. Time series of Landsat data were used to estimate hurricane damages of mangrove forest cover through December 31, 2017. We combined data from Landsat 7 ETM+ and Landsat 8 OLI to create a dense time series of cloud-free observations. All images were pre-processed to surface reflectance and masked for clouds using the same methods as the mangrove classification. Landsat 7 and Landsat 8 data were then harmonized to account for differences in the sensor specifications following[51]. We calculated the Normalized Difference Vegetation Index (NDVI) and Normalized Difference Water Index (NDWI) for each image in the collection. We calculated two reference maps from the time series of Landsat imagery (Supplementary Fig. 1). A pre-storm reference was calculated as the median value for each reflectance and index band for all cloud-free imagery in the two years prior Hurricane Irma, August 31, 2015 through August 31, 2017. Similarly, a post-storm median mosaic image was made using Landsat data between October 1, 2017 and December 31, 2017.

Pre- and post-storm wall-to-wall Fractional Vegetation Cover (FVC) maps were generated using a combination of lidar-based FVC metrics and Landsat imagery (Supplementary Fig. 1). First, lidar-based FVC was binned into five classes; 0–20%, 20–40%, 40–60%, 60–80%, and 80–100% (Supplementary Fig. 7). We then collected 1000 randomly generated points in each of the five FVC classes, a total of 5000 points, to be used as training data in the Landsat classification. Here, we implemented a Random Forest Classifier using 100 trees and a bag fraction of 0.5. These steps were applied to both the pre-storm and post-storm lidar-derived FVC and Landsat mosaic image metrics. Changes between the pre- and post-storm FVC were then calculated based on the five different FVC classes (Supplementary Fig. 7). For example, a pixel with pre-storm FVC of 80–100% and a post-storm FVC of 20–40%, a reduction of three FVC classes, was assigned a drop in FVC of 40–60% (Fig. 1).

**Recovery times and resilience**. We estimated the time to full recovery of pre-storm mangrove green canopy cover using the time series of Landsat NDVI during the first 15-months following Hurricane Irma. The pre-storm mean NDVI layer was used as a reference, as described in the previous section. Next we calculated the NDVI anomaly for each image during the post-storm period, September 17, 2017 through December 31, 2018 (Supplementary Fig. 1). We then summed the individual anomaly values from each Landsat image and normalized by the total number of valid pixels (i.e., pixels meeting quality control measures) to estimate the average change in NDVI within the 15 months after the storm. We used anomaly values to identify mangrove forests with large decreases in the 15 months after the

storm using a threshold of 0.2 for the 15-month NDVI average anomaly[19,52]. These areas suffered large losses of canopy material and limited new growth during the post-storm period. We used the slope in NDVI values for each pixel during 2018 to estimate the time in years to full recovery to pre-storm NDVI values, excluding data from October to December 2017 to remove delayed browning of damaged vegetation and spurious NDVI values from surface water features following the storm. Areas with a negative NDVI slope were not assigned a recovery time.

We used a combination of the NDVI slope, estimated time to full NDVI recovery, and the average change in NDVI between the pre- and post-storm periods to categorize mangrove forest resilience, the potential for mangroves to rebound to pre-disturbance conditions. The specific criteria for mangrove recovery rates and mangrove damage thresholds were adapted from field and remote sensing studies, respectively[6,19,25]. Regions of high resilience (a combination of high resistance and resilience) were identified based on rapid recovery and/or little to no immediate impact from the storm: (1) areas that were observed to recover to pre-disturbance conditions during 2018, (2) areas that were predicted to recover within 5 years regardless of the post-storm drop in NDVI[6], and (3) regions with a post-storm change in NDVI < −0.1, despite exposure to tropical storm or hurricane-force winds (Supplementary Fig. 3). The intermediate resilience class was classified as areas with predicted recovery times between 5 and 15 years[6] and areas with a negative NDVI slope or an extended recovery time but larger initial post-storm drop in NDVI of 0.1–0.2[25] (Supplementary Fig. 3). Lastly, the low resilience class of mangrove areas was defined as forests with predicted recovery times >15 years or a negative NDVI slope that occurred in regions with the largest (>0.2) post-storm drop in average NDVI[25] (Supplementary Fig. 9). The resilience class map is available online for download[53].

**Mangrove species and elevation**. We used species level maps developed by the National Park Service for Everglades National Park[50] to characterize the impact of Hurricane Irma on different mangrove species. For that study, dominant species were identified through photo-interpretation of stereoscopic, color-infrared aerial imagery. Grid cells of 50 m × 50 m covering an area (Region 3) of ~100,000 ha in southwest Florida were interpreted based on the majority cover type and validated using field observations. A total of 169 vegetation cover classes were identified in this region, however, only five mangrove cover classes were considered for these analyses: *Avicennia germinans* (Black Mangrove), *Laguncularia racemosa* (White Mangrove), *Rhizophora mangle* (Red Mangrove), *Conocarpus erectus* (Buttonwood), and a single mixed species mangrove class. Mangrove forest communities were defined as the dominant diagnostic species in the upper-most stratum[50]. The mangrove species data were reprojected to match the Landsat resolution and the resilience maps. We used the intersection of the resilience and species extent maps to estimate the proportion of each resilience class by dominant species.

The USGS National Elevation Dataset (NED) was used to estimate the soil elevation across southwest Florida. The 1/9 arc second (~3 m × 3 m) products were acquired from NED, and reprojected to Landsat resolution to estimate the proportion of each resilience class by soil elevation.

**Additional data and analysis**. Modeled maximum storm surge data for Hurricane Irma were acquired from Coastal Emergency Risks Assessment data portal. Storm surge is derived from the ADCIRC Prediction System that solves for time dependent, circulation, and transport in multiple dimensions[54]. Maximum sustained hurricane wind speed was modeled hourly at a 5 km × 5 km resolution for 2017 by NASA's Global Modeling and Assimilation Office (GMAO)[55]. The storm maximum wind speed for each 5 km × 5 km grid cell was calculated and binned into six discrete classes of wind speeds at 5 m s$^{-1}$ increments: 26–30, 31–35, 36–40, 41–45, 46–50, and >50.

**Statistical analyses**. Canopy height losses measured from NASA G-LiHT data were grouped by five pre-storm canopy height classes (0–5 m, 5–10 m, 10–15 m, 15–20 m, and >20 m). All valid pixels within the lidar footprint was used to calculate the mean, standard error, and area (sum of 1 m × 1 m pixels) for each class (Supplementary Table 1). These results were then tested for significant differences between canopy height losses and pre-storm canopy height classes between using a one-way ANOVA analysis with a post-hoc Tukey test in R (version 4.0.3). For testing the significance between environmental variables (i.e., pre-storm canopy height, canopy height loss, percent canopy height loss, surface elevation, and storm surge water level above ground) we employed a two-sided Kolmogorov–Smirnov test[56] implemented in R (version 4.0.3). First, we created a multi-band stacked image which included each of the variable layers. Within each resilience class (i.e., Low, Intermediate, and High) with randomly selected 10,000–20,000 points using Google Earth Engine to sample from the environmental variables images. From that sample set we then randomly selected 500 samples within each of the resilience classes. Each class combination (1) Low-Intermediate, (2) Low-High, and (3) Intermediate-High were compared using the Kolmogorov–Smirnov test. We repeated this procedure using 5000 iterations in order to provide a robust estimate of the Kolmogorov–Smirnov statistic, including the mean and first and third quartiles, which were then compared to the critical value (Supplementary Table 2).

**Reporting summary**. Further information on research design is available in the Nature Research Reporting Summary linked to this article.

## Data availability

All original datasets used in this study are freely or commercially available through their respective references and data portals, with the exception of the wind data. All original final data products generated and used in this study are archived at PANGAEA Data Publisher, https://doi.org/10.1594/PANGAEA.920522[53]. NASA G-LiHT canopy height and fractional vegetation cover data is available through the NASA G-LiHT web portal, https://glihtdata.gsfc.nasa.gov/. The Hurricane Irma maximum storm surge models are available at https://cera.coastalrisk.live/. The National Elevation Datasets are available at https://apps.nationalmap.gov/viewer/. NASA GMAO maximum wind data used is available upon reasonable request.

## Code availability

No new algorithms were developed in this study. Google Earth Engine code used to conduct the analysis is available from the corresponding author upon reasonable request.

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

## Acknowledgements

The authors thank Kimberly Rogers, Reide Corbett, Hannah, Jed Redwine, and Stephen Davis for providing edits on the manuscript. We thank Atticus Stovall assistance with the statistical analysis and Abigail Barenblitt with creating figures. DigitalGlobe data were provided by NASA's NGA Commercial Archive Data (cad4nasa.gsfc.nasa.gov) under the National Geospatial-Intelligence Agency's NextView license agreement. Funding: D.L. was supported by NASA's New Investigator Program (NNX16AK79G) and the Inter-agency Climate Change NASA program grant no. 2017-67003-26482/project accession no. 1012260 from the USDA National Institute of Food and Agriculture. D.L., T.F., B.D.C., L.A.C., and D.C.M. were supported by NASA's Rapid Response and Novel Ecosystem Studies Program (17-RRNES-0008). Partial funding to E.C.M was provided by the Florida Coastal Everglades Long-Term Ecological Research (FCE-LTER) program funded by the National Science Foundation (Grant #DEB-1832229). This is contribution number 1020 from the Southeast Environmental Research Center in the Institute of Environment at Florida International University. U.S. government sponsorship is acknowledged.

## Author contributions

D.L. and T.F. conceived and designed the study with input from D.C.M. D.L. led the execution of the analysis with contributions from T.F., D.C.M., and S.C. B.D.C. and L.A.C. collected and processed NASA G-LiHT data. P.M. and C.N. processed and provided WorldView stereo data. L.O. provided NASA GMAO wind data. D.L., T.F., D.C.M. and E.C.M. wrote the paper with input from all authors.

## Competing interests

The authors declare no competing interests.
