## [Peer Review File · Nature Communications]

Reviewer comments, first round

Reviewer #1 (Remarks to the Author):

In this study, the authors used remotely-sensed data to investigate the effects of Hurricane Irma on mangrove forests in Everglades National Park. The landscape-level data and results regarding mangrove recovery and structural effects of Hurricane Irma on mangrove forests in Everglades National Park are impressive and will provide a valuable foundation for future landscape-level assessments of hurricane impacts to mangrove forests and other coastal wetlands. It is an interesting, well-written, and valuable study. I enjoyed reading it! I do have a few suggestions for improving the communication of the results.

The manuscript has so much valuable information that the title and primary conclusions in the abstract may be overly simplistic given the diversity of information contained within the manuscript. Some of the most valuable results (e.g., damage to height classes, resilience metrics) could be given more weight in the title and abstract. From my perspective, the landscape-level resilience metrics are particularly valuable.

Is it possible that the mangrove forest mortality in some of the areas was already in progress before the Hurricane? Is it possible that some of the areas were impounded for many years (and perhaps decades) prior to Irma and were already in decline (e.g., the area near Flamingo) due to sustained flooding and anaerobic conditions? Then, Hurricane Irma delivered the final blow, which resulted in complete mortality? Some of the text refers to this (in the discussion), but this results would not be easily gleaned from the title and abstract.

Perhaps the title could be revised to: "Storm surge and hydrologic barriers contributed to mangrove dieback in southwest Florida following Hurricane Irma". Within that line of discussion, is there a way (at the landscape level) to quantify hydrologic impoundment and prior decline due to impoundment? Could you work to measure the extent of decline before the Hurricane in these poorly drained areas/hydrologic impoundments?

L25- Replace "triggered" with "contributed to" because Irma didn't begin the dieback. These were areas that were stressed before the hurricane.

L29- There are important restoration and management implications here that could be stated in the concluding sentence. Tidal restoration and hydrologic management can improve mangrove resilience and reduce mortality.

L55- Here, I suggest also adding some mention of prior condition, especially within the context of prior alterations to hydrologic regimes and tidal connectivity. See Lewis et al. 2016 and some of the reference noted below (Bashan 2013; Vogt 2012; Harris 2010).

L129- check this sentence

L193- this point (and the management implications) could be expanded upon and highlighted. It's possible that something could have been done to improve the hydrology in these areas, which could have improved resilience and reduced the extent of mortality.

L277- It's possible that many of these forests that experienced dieback were already in decline for many years before the storm (e.g., the area near Flamingo has been stressed for many years prior to Irma). It would have been interesting to quantify this in more detail.

Fig. S9- perhaps "High resilience" should be replaced with "High resistance" (Fig. S9C)

Here are some additional relevant papers that may be of interest:

Bashan, Y., M. Moreno, B. G. Salazar, and L. Alvarez. 2013. Restoration and recovery of hurricane-damaged mangroves using the knickpoint retreat effect and tides as dredging tools. *Journal of Environmental Management* 116:196-203.

Bischof, B. G. 1995. Aerial photographic analysis of coastal and estuarine mangrove system dynamics of the Everglades National Park, Florida, in response to hurricanes: implications for the continuing sea-level rise. M.S. Thesis. University of Miami. Coral Gables, Florida, USA.

Chambers, L. G., H. E. Steinmuller, and J. L. Breithaupt. 2019. Toward a mechanistic understanding of "peat collapse" and its potential contribution to coastal wetland loss. *Ecology* 100:e02720.

Craighead, F. C. and V. C. Gilbert. 1962. The effects of Hurricane Donna on the vegetation of southern Florida. *Quarterly Journal of the Florida Academy of Sciences* 25:1-28.

Craighead, F. C. 1964. Land, mangroves and hurricanes. *The Fairchild Tropical Garden Bulletin* 19:1-28.

Harris, R. J., E. C. Milbrandt, E. M. Everham, and B. D. Bovard. 2010. The effects of reduced tidal flushing on mangrove structure and function across a disturbance gradient. *Estuaries and Coasts* 33:1176-1185.

Lewis III, R. R., E. C. Milbrandt, B. Brown, K. W. Krauss, A. S. Rovai, J. W. Beever, and L. L. Flynn. 2016. Stress in mangrove forests: Early detection and preemptive rehabilitation are essential for future successful worldwide mangrove forest management. *Marine Pollution Bulletin* 109:764-771.

Osland, M. J., L. C. Feher, G. H. Anderson, W. C. Vervaeke, K. W. Krauss, K. R. T. Whelan, K. M. Balentine, G. Tiling-Range, T. J. Smith III, and D. R. Cahoon. 2020. A tropical cyclone-induced ecological regime shift: mangrove conversion to mudflat in Florida's Everglades National Park (Florida, USA). *Wetlands* <https://doi.org/10.1007/s13157-020-01291-8>.

Sippo, J. Z., C. E. Lovelock, I. R. Santos, C. J. Sanders, and D. T. Maher. 2018. Mangrove mortality in a changing climate: An overview. *Estuarine, Coastal and Shelf Science* 215:241-249.

Vogt, J., A. Skóra, I. C. Feller, C. Piou, G. Coldren, and U. Berger. 2012. Investigating the role of impoundment and forest structure on the resistance and resilience of mangrove forests to hurricanes. *Aquatic Botany* 97:24-29.

Wanless, H. R. and B. M. Vlaswinkel. 2005. Coastal landscape and channel evolution affecting critical habitats at Cape Sable, Everglades National Park, Florida. Final Report to Everglades National Park. University of Miami, Coral Gables, Florida, USA.

Reviewer #2 (Remarks to the Author):

"Storm surge, not wind caused mangrove dieback in southwest Florida following Hurricane Irma".

General comments:

This is a very interesting paper, and I believe its novelty lies in that the question of the dominant cause of mangrove dieback has not been attacked with such intensive remote sensing before. The alignment of differing data sources into a story is generally powerful. I believe that the methods used do indeed lead to the conclusions stated. In this way, I believe the paper will eventually be of high interest to readers. However, I believe the presentation to be a little overly complex for a short communication and could be improved.

In particular, after reading an excellent introduction, I was troubled with the clarity and flow of the Results and Discussion sections. The two seemed to be somewhat jumbled. In addition, there was a high number of figures that I felt were underutilized. Because of this, the authors introduced confusion took some punch away from the context and importance of the key findings that I expected in the Discussion. Regardless, I think the story is strong, and with some trimming and revising (specific suggestions below), this paper will ultimately be highly suitable for *Nature Communications*.

Specific comments:

Results: I would like to see a little more description here to help readers track which parts of the methods are leading to specific results. In this report format, the reader knows little of the methods yet except for the brief overview in the final paragraph of the introduction. This does not have to be wordy, but just a few words to clue in readers to which methods to check later. For instance, Line 96 "Structural damages to mangrove forests" could be "Structural damages to mangrove forests, as measured by pre and post storm satellite and LiDAR canopy height models, ..."

Line 102, how is volume measured here, there is no reference to "volume" in methods.
Line 103, More details in the spatial patterns here would help keep it out of the discussion later.

Overall, this results seems excessively brief, given the abundance of spatial information the authors created. All I see in here are survey area summaries. There are some sentences in the discussion that should be in this section. Having all the results in one place will help prepare for what will be discussed later, and the discussion would not seem as interleaved.

Discussion:

I would like to see more context of how these results are new and important and about spatial patterns that bring the results into unique features of the landscape. There's also no real powerful paragraph where the main finding is highlighted (closest is line 170, at end of a paragraph in the middle of the discussion)

Line 125: If spatial patterns were laid out in results, the authors could be specific here.

Lines 140-146: This needs to be in the Results section.

Lines 155-160: This is pretty dry results-section stuff.

Line 161: Great context paragraph.

Line 163: I think "in terms of" instead of "based on" sounds better.

Line 184-: Great paragraph on how results are useful in the future.

Can the authors discuss how the stereo pair estimation might be affected by mangrove/understory species and whether this would affect canopy loss results.

Methods: Pretty clear and concise, though I have some thoughts on fig S1 further down.

Figures:

A lot of figures with some redundancy and an order that does not entirely follow their natural order in the methods / results. Here are some thoughts to reduce and maximize effect of each figure.

Fig 1. This is busy, and not actually referred to that much. I would like panels D,E, and F alone. The ground photos don't seem necessary since they are not really tied to text, nearly duplicated in fig S5. If the authors do keep them, they should be described more in the caption.

Fig 2. Looks neat, but I'm not sure it is used effectively. I think it should be introduced first in the results, where the spatial patterning is described. Then this spatial patterning of resilience can be discussed in the discussion. Also, can the authors fit a brief definition of resilience levels here?

Fig 3. Not entirely sure this is needed.

Fig S1. If the authors could simplify this or break it into sections that could be referred to in the text of the methods, then I think this would help track where the different responses are coming from.

Fig S2. This info could be combined with panels A,B,and C of figure 1 into a separate figure.

Fig S3, S4, S5. Keep these.

Fig S6. Not sure this needs to be a figure, when the regression stats tell pretty much the same story.

Fig S7. This is not really helpful and should be dropped.

Fig S8. This figure should be in the main section of the paper. Perhaps with another figure of the height loss with the photos of each Can the authors give a better timestamp for the post-Irma FVC map?

Fig S9. Yes, these clarify the resilience methods. Can NDVI slope be added to these to better show how recovery time is estimated?

Table S1. Keep this.

I do hope the editors and authors find my thoughts to be constructive and beneficial.

Signed,

Dr. Nick Vaughn

Center for Global Discovery and Conservation Science
Arizona State University

Reviewer #3 (Remarks to the Author):

The manuscript by Lagomasino claims that storm surge, not wind, caused mangrove dieback in SW Florida following Hurricane Irma however additional information is required to support this conclusion. The first sentence of the results section states that structural damage to mangrove forests from Irma varied based on wind exposure and canopy height. Although the authors provided a table showing 'change in mean canopy height' by 'wind speed class', no statistical methods or results were included in the manuscript to support interpretation of the table or findings. The authors state that damage varies with canopy height, but it is unclear if taller canopies suffer proportionally more damage than shorter canopies or if any of the observed differences are statistically significant. This section needs to be strengthened to better make the case that wind did not cause dieback. The authors also stated that areas of greatest fractional vegetation cover (FVC) loss were not co-located with the largest reductions in canopy height except where Irma made landfall but it is not clear what the relationship is between damage and dieback.

The crux of the article is that storm surge and not wind caused mangrove dieback however the results section focuses exclusively on the relationship between wind and immediate post storm damage (and not dieback or recovery). The authors do mention in the results section that the greatest losses in canopy height and volume were concentrated in the major estuaries where regular flushing takes place, but this does not provide evidence of storm surge being the primary driver of dieback. The results section does not quantify or address losses in canopy cover or canopy height with respect to storm surge, elevation or any other proxy for storm surge. Most of key results are in the discussion section (Figure 2) and need to be moved to the results section. Figure 2 begins to address the main hypothesis of the paper by comparing die back (low resilience) to elevation. Again, this is still not directly relating die back to storm surge but provides information about the relationship between elevation and mangrove resilience to cyclones. The authors again reference that delayed or failed mangrove recovery was primarily confined to endorheic basins but this argument would be stronger with a spatial analysis and quantitative results depicting the portion of mangroves from each resilience class that are located within predefined endorheic basins. The authors conclude that mangrove die-off resulted from the retention of storm surge water in enclosed or semi-enclosed basins. This manuscript would be strengthened by including and analyzing spatial information showing these surge retention areas

(remotely sensed or field collected) or the barriers that create them.

Once the resilience (dieback) results are moved into the results section, the discussion can focus (as it currently does) on possible explanations of the dieback hotspots. The discussion does a nice job highlighting the management and research implications of the findings.

I recommend this paper be accepted pending major revisions to provide additional statistical and geospatial data to support the conclusions drawn by the authors. As the authors state, mangroves reduce risk to coastal communities and additional science that can inform on mangrove recovery is needed to ensure that mangroves can continue to provide this critical ecosystem service in the face of increasing frequency and intensity of tropical cyclones. This paper presents innovative and comprehensive methods for assessing mangrove recovery and with some additional data and analysis could provide much needed insights on how hydrologic alterations are reducing the resilience of mangroves to storms and in turn, reducing the benefits they provide to coastal communities.

Author Response to Reviewer 1 - NCOMMS-20-23766A

In this study, the authors used remotely-sensed data to investigate the effects of Hurricane Irma on mangrove forests in Everglades National Park. The landscape-level data and results regarding mangrove recovery and structural effects of Hurricane Irma on mangrove forests in Everglades National Park are impressive and will provide a valuable foundation for future landscape-level assessments of hurricane impacts to mangrove forests and other coastal wetlands. It is an interesting, well-written, and valuable study. I enjoyed reading it! I do have a few suggestions for improving the communication of the results.

We are happy to hear the reviewer's thoughts on our manuscript, and that it was an enjoyable and informing read. Thank you.

The manuscript has so much valuable information that the title and primary conclusions in the abstract may be overly simplistic given the diversity of information contained within the manuscript. Some of the most valuable results (e.g., damage to height classes, resilience metrics) could be given more weight in the title and abstract. From my perspective, the landscape-level resilience metrics are particularly valuable.

We thank the reviewer for the kind remarks on our results. We have revised the title slightly to capture the comments made by multiple reviewers. Similarly, we have added to the abstract to highlight aspects of the environmental setting. Lines 27-29

In addition, we have revised several sections in the manuscript to highlight these points further. Please see marked-up draft.

Is it possible that the mangrove forest mortality in some of the areas was already in progress before the Hurricane? Is it possible that some of the areas were impounded for many years (and perhaps decades) prior to Irma and were already in decline (e.g., the area near Flamingo) due to sustained flooding and anaerobic conditions? Then, Hurricane Irma delivered the final blow, which resulted in complete mortality? Some of the text refers to this (in the discussion), but this results would not be easily gleaned from the title and abstract.

These are very important comments, and ideas that the author's have thought about. The reviewer lists a number of circumstances that could have ultimately lead to the high mortality areas. We followed up with an analysis of pre- and post-spectral signatures from Landsat satellite data. There was not any statistical difference between the spectral reflectance data for the different resilience classes prior to Hurricane Irma. We agree that Irma could have been final blow to areas that were already degraded, and areas that the reviewer mentioned did show degradation on the ground prior to the storm. We believe it would be prudent to investigate this in more detail as a follow up analysis. However, we have revised the text to better highlight the potential for "pre-existing" conditions.

Lines 63, 213-215

Perhaps the title could be revised to: “Storm surge and hydrologic barriers contributed to mangrove dieback in southwest Florida following Hurricane Irma”. Within that line of discussion, is there a way (at the landscape level) to quantify hydrologic impoundment and prior decline due to impoundment? Could you work to measure the extent of decline before the Hurricane in these poorly drained areas/hydrologic impoundments?

Please see our previous response on pre-storm declines. The reviewer brings up some good points. These are results of the work that came out after our analysis. We conducted an spectral analysis between resilience classes before the storm and we did not measure any significant differences. As we stated earlier, a more detailed analysis of the pre-storm condition would be important for a follow up investigations.

L25- Replace “triggered” with “contributed to” because Irma didn’t begin the dieback. These were areas that were stressed before the hurricane.

We agree that some, if not most, of these areas were stressed before the Irma hit. The word triggered is used here to identify that it was the final process that lead to the dieoff. The area would have been stressed, but if conditions changes (e.g., drought relief, flood relief) the area may have rebounded. Irma may have just pushed it beyond its limits. From our analysis of the pre-storm Landsat image mosaic there was no significant differences in spectral data between resilience classes. Please see our previous comment. We keep triggered in the text, but have revised the text to better highlight pre-existing conditions

L29- There are important restoration and management implications here that could be stated in the concluding sentence. Tidal restoration and hydrologic management can improve mangrove resilience and reduce mortality.

These are important points. We have made revisions to the last paragraph to address these comments. Lines 31-32

L55- Here, I suggest also adding some mention of prior condition, especially within the context of prior alterations to hydrologic regimes and tidal connectivity. See Lewis et al. 2016 and some of the reference noted below (Bashan 2013; Vogt 2012; Harris 2010).

We agree with the reviewer that prior condition should also be added in this context. Revisions have been made in the text to reflect this point.

Line 63

L129- check this sentence

We are not sure what the reviewer was referring to here. However, we have split this sentence to ensure readability. Lines 192-194

L193- this point (and the management implications) could be expanded upon and highlighted. It's possible that something could have been done to improve the hydrology in these areas, which could have improved resilience and reduced the extent of mortality.

We have made several revisions to these sentences to better highlight the role of remote sensing to identify the basin and other stress indicators and how remediation to the hydrology will help to reduce stress.

Lines 250-253

L277- It's possible that many of these forests that experienced dieback were already in decline for many years before the storm (e.g., the area near Flamingo has been stressed for many years prior to Irma). It would have been interesting to quantify this in more detail.

We agree with the reviewer comments. We acknowledge that pre-existing conditions may have been a primer that when combined with Irma could have lead exaerbated the damage. However, quantifying those prior conditions were not within the scope here, but a good point to focus on in follow on work. Our major focus here was to identify those specific regions that had issues and understand the structural components. Indeed, prior conditions will be our next step. We have made revisions in the main text to highlight the importance of these onset conditions.

Fig. S9- perhaps "High resilience" should be replaced with "High resistance" (Fig. S9C)

We agree with the reviewer regarding the defining this class. However, high resistance does not capture this class as there are also aspects of recovery time that we include in our definition. We have updated this section to reflect the definition of the high resilience class to also include high resistance. This class is measured by the combination of little to no impact (high resistance - rule 3) and recovery time (high resilience - rule 1 & 2).

Lines 374-375

Here are some additional relevant papers that may be of interest:

Thank you for the reference list. On recommendations made in the previous comments we have included a few additional references from this list in the manuscript that help put into context mangrove rehabilitation and ecological shifts.

Author Response to Reviewer 2 - NCOMMS-20-23766A

This is a very interesting paper, and I believe its novelty lies in that the question of the dominant cause of mangrove dieback has not been attacked with such intensive remote sensing before. The alignment of differing data sources into a story is generally powerful. I believe that the methods used do indeed lead to the conclusions stated. In this way, I believe the paper will eventually be of high interest to readers. However, I believe the presentation to be a little overly complex for a short communication and could be improved.

We appreciate the comments from the reviewer. We are happy that our overall framework resonated with the reviewer. Our revisions to the manuscript based on these comments and those from the other reviewers will hopefully address the concerns.

In particular, after reading an excellent introduction, I was troubled with the clarity and flow of the Results and Discussion sections. The two seemed to be somewhat jumbled. In addition, there was a high number of figures that I felt were underutilized. Because of this, the authors introduced confusion took some punch away from the context and importance of the key findings that I expected in the Discussion. Regardless, I think the story is strong, and with some trimming and revising (specific suggestions below), this paper will ultimately be highly suitable for Nature Communications.

We thank the review for their insight here regarding the organization and underutilization of some figures. We address these comments more specifically later, however we have made several revisions to the manuscript and modified a few figures to better highlight the results of our study.

Results

Introduction. This does not have to be wordy, but just a few words to clue in readers to which methods to check later. For instance, Line 96 "Structural damages to mangrove forests" could be "Structural damages to mangrove forests, as measured by pre and post storm satellite and LiDAR canopy height models, ..."

We agree with the reviewer here about adding a bit more detail to the sentence to connect the reader with the method.

Lines 138-140

Revised: Where mortality and recovery were closely related to soil elevation and storm surge, structural damages to mangrove forests caused by Hurricane Irma, as measured by the change in satellite and lidar canopy height models, varied based on wind exposure and pre-storm canopy height (Fig. 1, fig. S4 and S5).

Line 102, how is volume measured here, there is no reference to "volume" in methods.

This was an oversight that we left out. We thank the reviewer for making note of this issue. We have made revisions to address the missing methodology.

Line 304-307

Line 103, More details in the spatial patterns here would help keep it out of the discussion later.

We have many revisions to the text to better highlight key spatial patterns, particularly with flooded regions. Please see marked-up draft - Results section and new supplemental figures.

Overall, this results seems excessively brief, given the abundance of spatial information the authors created. All I see in here are survey area summaries. There are some sentences in the discussion that should be in this section. Having all the results in one place will help prepare for what will be discussed later, and the discussion would not seem as interleaved.

We have revised the results to reflect the comments of the Reviewer as well as those from a separate Reviewer. The results sections has been realigned to focus on the storm surge and location environmental patterns in more detail. The figures have been revised following reviewer recommendations and additional analyses.

See marked up draft for revisions.

Discussion

I would like to see more context of how these results are new and important and about spatial patterns that bring the results into unique features of the landscape. There's also no real powerful paragraph where the main finding is highlighted (closest is line 170, at end of a paragraph in the middle of the discussion)

We thank the reviewer of their comments here. Using these comments we restructure the results to better focus on spatial patterns and revised this paragraph to better highlight the novel results from this research.

Line 125: If spatial patterns were laid out in results, the authors could be specific here.

Revised and restructured the Results. See marked-up draft

Lines 140-146: This needs to be in the Results section.

Revised and restructured the Results. See marked-up draft

Lines 155-160: This is pretty dry results-section stuff.

Considering the reviewer comments we have revised and restructured the results section with a bit more gusto. See marked-up draft

Line 161: Great context paragraph.

Thank you

Line 163: I think "in terms of" instead of "based on" sounds better.

We have made the suggested revisions.

Line 218

Line 184-: Great paragraph on how results are useful in the future.

Thank you. We appreciate your comments on this paragraph. This when through numerous iterations. We will also mention, that we revised the concluding paragraph slightly to address comments from other reviewers.

Can the authors discuss how the stereo pair estimation might be affected by mangrove/understory species and whether this would affect canopy loss results.

The Digital Surface Model derived from the satellite stereo pair represents the highest resolvable canopy surface. In closed canopy surface models, like the case for the Everglades, understory characteristics do not have an impact on deriving this DSM.

Methods

Pretty clear and concise, though I have some thoughts on fig S1 further down.

Thank you, we appreciate the comments on the methods. We have made some revisions based on some of the comments on the figures and the main text - just for additional clarity.

Figures

A lot of figures with some redundancy and an order that does not entirely follow their natural order in the methods / results. Here are some thoughts to reduce and maximize effect of each figure.

We appreciate the comments here and have addressed them in various ways, including removing, reorganizing, and referencing the figures. Please see more detailed responses below.

Fig 1. This is busy, and not actually referred to that much. I would like panels D,E, and F alone. The ground photos don't seem necessary since they are not really tied to text, nearly duplicated in fig S5. If the authors do keep them, they should be described more in the caption.

We appreciate the review comments. We have simplified Fig 1 to better highlight the datasets and field conditions. We made sure to explicitly refer to each component in the text, including the photos, and we also provided additional details in the captions.

Fig 2. Looks neat, but I'm not sure it is used effectively. I think it should be introduced first in the results, where the spatial patterning is described. Then this spatial patterning of resilience can be discussed in the discussion. Also, can the authors fit a brief definition of resilience levels here?

We thank the review for their comments. We have revised the text and figures. Please see previous comments. Definitions are provided in the Supplemental, however, we agree that the classes should be briefly defined in the main text. We have made revisions to address the reviewer comments. See marked-up draft - revised Results section.

Fig 3. Not entirely sure this is needed.

We appreciated the reviewer's comment. However, we believe that this figure is useful for generalizing the environmental settings for each of the cover types and is key to summarizing our findings.

Fig S1. If the authors could simplify this or break it into sections that could be referred to in the text of the methods, then I think this would help track where the different responses are coming from.

We understand that this at first look, might be a complicated figure. We have made a few revisions and reordered the figure that seeks to better define each step of the model. We have also edited the caption to help better convey the methods framework.

Fig S2. This info could be combined with panels A,B,and C of figure 1 into a separate figure.

We agree with the reviewer. Fig S2 has been combined with Figure 1.

Fig S3, S4, S5. Keep these.

Great. Thank you.

Fig S6. Not sure this needs to be a figure, when the regression stats tell pretty much the same story.

We appreciate the comments from the reviewer. However, we do insist on these figures remaining in the text. These figures are used to highlight the high-resolution stereo datasets that were used in the canopy height and forest volume analysis. We want to highlight the use and capabilities of spaceborne canopy structure models for purposes like these to help augment airborne studies. We have made several revisions throughout the doc to better refer to the figure.

Fig S7. This is not really helpful and should be dropped.

Please see our previous response

Fig S8. This figure should be in the main section of the paper. Perhaps with another figure of the height loss with the photos of each Can the authors give a better timestamp for the post-Irma FVC map?

Thank you for the suggestion regarding the figures. We agree that this is a helpful figure. We included the change image (pre minus post) in the Figure 1, as we believe the change summarizes the canopy fraction loss succinctly. We have made small revisions to the text to better highlight the change as a well as the pre and pre maps by using this figure reference.

Fig S9. Yes, these clarify the resilience methods. Can NDVI slope be added to these to better show how recovery time is estimated?

This is now Fig. S3. Slopes have been added to the picture

Author Response to Reviewer 3 - NCOMMS-20-23766A

The manuscript by Lagomasino claims that storm surge, not wind, caused mangrove dieback in SW Florida following Hurricane Irma however additional information is required to support this conclusion. The first sentence of the results section states that structural damage to mangrove forests from Irma varied based on wind exposure and canopy height. Although the authors provided a table showing 'change in mean canopy height' by 'wind speed class', no statistical methods or results were included in the manuscript to support interpretation of the table or findings. The authors state that damage varies with canopy height, but it is unclear if taller canopies suffer proportionally more damage than shorter canopies or if any of the observed differences are statistically significant. This section needs to be strengthened to better make the case that wind did not cause dieback. The authors also stated that areas of greatest fractional vegetation cover (FVC) loss were not co-located with the largest reductions in canopy height except where Irma made landfall but it is not clear what the relationship is between damage and dieback.

We appreciate the reviewers candid remarks on our manuscript. These comments, in addition to those from the other reviewers, have been extremely helpful in reorganizing our paper to better highlight the impacts of hurricanes on mangroves at the landscape scale. We have provided additional statistical tests to address the reviewer concerns. Wind and height categories were tested for separability through a paired ANOVA and post-hoc tukey test (see revised Table 1 and Methods section). The results from that analysis show significant differences between each of the canopy height classes. Though the standard deviations are high, due to the high number of observations, the standard error is extremely low. The standard error and the number of observations (reported as G-LiHT areal extent) are reported in Table S1. For this study, we focused only on the total height loss which provide information on severe damage rather than the percent loss, as this value better represents the potential biomass loss to the system. To help support this further, within each resilience class we randomly sampled points to extract environmental and canopy structure information (see fig. S4, table S1, and Statistical Analysis Section (Lines 413-430). Here we applied a Kolmogorov-Smirnov test to determine the difference in class distributions using 5000 iterations of 500 random samples, the distributions of the KS stat and the KS critical value are given in table S1.

We also provided additional analysis on the distribution of FVC loss and resilience class (see new fig. S8). Here we show that all FVC loss classes had the dieback (low resilience) class, whereas the most severe canopy cover loss class overlapped primarily with the dieoff class. See accompanying text Lines 151-152 and 176-179.

We made several revisions throughout the document to address the review concerns and highlight the additional analyses.

The crux of the article is that storm surge and not wind caused mangrove die back however the results section focuses exclusively on the relationship between wind and immediate post storm damage (and not dieback or recovery). The authors do mention in the results section that the greatest losses in canopy height and volume were concentrated in the major estuaries where regular flushing takes place, but this does not provide evidence of storm surge being the primary driver of dieback. The results section does not quantify or address losses in canopy cover or canopy height with respect to storm surge, elevation or any other proxy for storm surge.

Based on the reviewer concerns, we have provided additional analysis that further supports the role of storm surge impacts on mangrove mortality (new Fig. 1C) and analyzed the impact of canopy cover loss (new fig. S8), and analyzed the distribution of pre-storm canopy height, canopy height loss, elevation, and storm surge above ground (fig. S4 and table S1).

Most of key results are in the discussion section (Figure 2) and need to be moved to the results section. Figure 2 begins to address the main hypothesis of the paper by comparing die back (low resilience) to elevation. Again, this is still not directly relating die back to storm surge but provides information about the relationship between elevation and mangrove resilience to cyclones. The authors again reference that delayed or failed mangrove recovery was primarily confined to endorheic basins but this argument would be stronger with a spatial analysis and quantitative results depicting the portion of mangroves from each resilience class that are located within predefined endorheic basins. The authors conclude that mangrove die-off resulted from the retention of storm surge water in enclosed or semi-enclosed basins. This manuscript would be strengthened by including and analyzing spatial information showing these surge retention areas (remotely sensed or field collected) or the barriers that create them.

The reviewer has provided some helpful comments here. We have moved figure 2 to the results section. We have also reorganized the results section to highlight the additional analysis and present the dieback, elevation, and storm surge impacts first then followed by the impacts to canopy structure. For the analysis, we tested the differences in storm surge as well as canopy height loss for each resilience class (see Statistical Analysis section). We have added an additional figure panel showing the relationship between resilience class and the max surge water level above the ground surface (Fig. 2C). In addition, we have made a number of revisions throughout the document to address the reviewer concerns. Please see marked up draft for all revisions.

We agree that a detailed spatial analysis of the watersheds of the Everglades would help to better delineate the endorheic basins. However, the microtopography of the region can make this analysis quite challenging. Though topographic highs and lows can be identified with the data, small variations in ground elevation can drastically impact the drainage modeling. This type of microtopography watershed analysis warrants its own detailed study. In fact, the results of our study may help in helping delineate microtopographic basins. We have highlighted these needs in Lines 250-253

Once the resilience (dieback) results are moved into the results section, the discussion can focus (as it currently does) on possible explanations of the dieback hotspots. The discussion does a nice job highlighting the management and research implications of the findings.

We thank the reviewer for their comments on this section. We have moved the resilience paragraph to the beginning of the results as well as provided additional information. Please see comments above and the marked-up draft, for a full view of revisions.

I recommend this paper be accepted pending major revisions to provide additional statistical and geospatial data to support the conclusions drawn by the authors. As the authors state, mangroves reduce risk to coastal communities and additional science that can inform on mangrove recovery is needed to ensure that mangroves can continue to provide this critical ecosystem service in the face of increasing frequency and intensity of tropical cyclones. This paper presents innovative and comprehensive methods for assessing mangrove recovery and with some additional data and analysis could provide much needed insights on how hydrologic alterations are reducing the resilience of mangroves to storms and in turn, reducing the benefits they provide to coastal communities.

Again, we would like to thank the review for their time reviewing and commenting on our manuscript. These comments have been extremely helpful in focusing on the major results of our study, as well as providing additional statistical support to our conclusions. Some of the suggested analysis was, in part, out of the scope of this particular work, we recognize the needs for these analyses. We have addressed some of these needs in the final paragraph, as delineating this basins will be important managment efforts.

Reviewer comments, second round –

Reviewer #1 (Remarks to the Author):

The authors have improved the manuscript and addressed my concerns. I've included a few additional comments below.

L47- check spelling of phosphorus (not phosphorous)

L81- should fig be upper case?

L125 and L195- I'm a little concerned by the focus on *A. germinans* as a species here and the message that it conveys to readers. Somewhere in here, I think it would be good to make it clear that many *A. germinans* individuals recovered in the blue and green areas on the map (my personal observation). So, it's not a species-specific effect but more so reflective of the landscape position of forests where *A. germinans* is dominant (basin forests perhaps?). Within mixed forest that contain all three species, was *A. germinans* also disproportionately affected? Field-based observations made by the coauthors within the LTER sites could serve as a good check for the focus on *A. germinans*. Is this what plant ecologists working on the ground have also seen in Irma-damaged vegetation plots? If not, then it may be best to focus more of the text on landscape position and less on the species. Or some qualifying text could be added.

Table 1- the caption and table should make it clear that #s within the table refer to canopy height reductions (in m). For consistency- use mean rather than average.

L621- check this sentence

L184- add citation to support this statement (i.e., after "can take decades.")

L188- replace "post-" with following

L195- see prior comment regarding the species focus

L200- here the landscape position (i.e., basin forests) are introduced and emphasized

L219- clarify which hurricane this statement refers to (Donna or Irma)

L216- might be worth briefly mentioning the 1935 Labor Day Hurricane here too, as it led to mass mangrove mortality

Reviewer #3 (Remarks to the Author):

My concerns have been addressed in the revisions.

Author's response to final reviewer comments -

L47- check spelling of phosphorus (not phosphorous)

Corrected

L81- should fig be upper case?

Corrected throughout to comply with editorial requests.

L125 and L195- I'm a little concerned by the focus on *A. germinans* as a species here and the message that it conveys to readers. Somewhere in here, I think it would be good to make it clear that many *A. germinans* individuals recovered in the blue and green areas on the map (my personal observation). So, it's not a species-specific effect but more so reflective of the landscape position of forests where *A. germinans* is dominant (basin forests perhaps?). Within mixed forest that contain all three species, was *A. germinans* also disproportionately affected? Field-based observations made by the coauthors within the LTER sites could serve as a good check for the focus on *A. germinans*. Is this what plant ecologists working on the ground have also seen in Irma-damaged vegetation plots? If not, then it may be best to focus more of the text on landscape position and less on the species. Or some qualifying text could be added.

We thank the reviewer for the additional comments and considerations regarding species-effects. We do agree with the author that not all *A. germinans* were impacted by the storm, it was in fact, *A. germinans* dominant forest stands that were most impacted. Similarly we do agree that it is landscape position that is the important driver of vulnerability. However, we still do want to highlight these forest community traits since species composition can be more readily detected from surveys and remote sensing compared to elevation. We have revised several sections in the main text and methods to highlight the forest stand versus individual trees as well as further highlight the role of elevation and connectivity.

Lines 111-115: Dieback was also disproportionally concentrated in forest stands dominated by *Avicennia germinans* with an estimated impact area of 73% (7,750 ha) of all dieback areas (Fig. 2d). Overall, Irma had a strong selective pressure on the distribution of forests dominated by *A. germinans* whereby nearly 40% of these mangrove forests died compared to the less than 6% of the other forest communities.

Lines 180-184: We found greater vulnerability of monospecific and co-dominated stands of *A. germinans* to dieback following Hurricane Irma than other species communities. However, this species-specific vulnerability may reflect the competitive exclusion of other species in low-lying or hydrologically isolated areas prior to the storm, as these basin mangrove ecotypes are generally dominated or co-dominated by *A. germinans* [1], the most salt-tolerant species in the neotropics.

Table 1- the caption and table should make it clear that #s within the table refer to canopy height reductions (in m). For consistency- use mean rather than average.

Corrected

L621- check this sentence

We are not sure what the reviewer was referring to in this comment. We have reviewed the sentence but issues were not apparent

L184- add citation to support this statement (i.e., after “can take decades.”)

Added the Danielson et al, 2017 reference

L188- replace “post-“ with following

Corrected

L195- see prior comment regarding the species focus

Please see prior response to comment

L200- here the landscape position (i.e., basin forests) are introduced and emphasized

Please see prior response to comment

L219- clarify which hurricane this statement refers to (Donna or Irma)

We have clarified the statement

L216- might be worth briefly mentioning the 1935 Labor Day Hurricane here too, as it led to mass mangrove mortality

We have added some discussion regarding the Labor Day storm.

Lines 193-199: The south-to-north track of Hurricane Irma resulted in greater mangrove dieback than other recent hurricanes to strike southwest Florida, including Category 4 or Category 5 Hurricanes Charley (2005) and Andrew (1992), respectively [2,3]. The last two major storms with a similar trajectory over the Florida Keys and Florida Bay, Hurricane Donna (1960, Category 4) and the Labor Day

Hurricane (1935, Category 5) [4], also triggered widespread damage and the eventual collapse of coastal forests [5]. Unlike Irma, little is known about the total extent of dieback from Donna or the Labor Day Hurricane.